# Limits on determining the skill of North Atlantic Ocean decadal predictions

Matthew B. Menary [1] & Leon Hermanson [1]

The northern North Atlantic is important globally both through its impact on the Atlantic Meridional Overturning Circulation (AMOC) and through widespread atmospheric tele-connections. The region has been shown to be potentially predictable a decade ahead with the skill of decadal predictions assessed against reanalyses of the ocean state. Here, we show that the prediction skill in this region is strongly dependent on the choice of reanalysis used for validation, and describe the causes. Multiannual skill in key metrics such as Labrador Sea density and the AMOC depends on more than simply the choice of the prediction model. Instead, this skill is related to the similarity between the nature of interannual density variability in the underlying climate model and the chosen reanalysis. The climate models used in these decadal predictions are also used in climate projections, which raises questions about the sensitivity of these projections to the models' innate North Atlantic density variability.

[1] Met Office Hadley Centre, Met Office, Exeter EX1 3PB, UK. Correspondence and requests for materials should be addressed to M.B.M. (email: matthew.menary@metoffice.gov.uk)

Following record warm years in 2014 and 2015, the year 2016 was again the warmest year in the instrumental record and likely the warmest in at least the last one hundred thousand years[1–3]. Despite this, there has been well-documented recent cooling in the North Atlantic subpolar gyre (NA SPG)[4]. Indeed, the climate model simulations of the coming century project a weaker warming in the NA SPG compared to the global mean[5], highlighting the potential regional differences in the patterns of future climate change[6].

In addition, the NA SPG is of particular interest as it represents an important part of the widely studied Atlantic Multidecadal Oscillation/Variability (AMO/V)[7–11], which has been linked with a variety of climate phenomena, such as tropical storms, droughts in Africa, and summertime climate over Europe[12–14]. As such, it is both a region that may be experiencing important shifts and one in which these shifts may have significant climatic impact.

The NA SPG has also been the focus of study in initialised decadal climate predictions. It is a region that may actually be predictable up to a decade ahead[15], and good initialisation of the NA SPG has been shown to be important in providing potential predictability in other regions, such as in decadal forecasts of tropical storms[16]. These initialised predictions come in two main types that differ essentially on whether the model biases are removed before (anomaly method) or after (full-field method) the forecast is created[17]. Within the NA SPG, the Labrador Sea is of particular importance as it links the circulation in the surface and the deep ocean, contributing to the lower limb of the upper cell of the Atlantic Meridional Overturning Circulation (AMOC)[18,19].

If these decadal predictions are to be useful, they must be shown to have skill when forecasting the real world ocean, which requires producing re-forecasts (hindcasts) over many historical decades[20]. In the absence of spatially complete observations stretching back many decades, ocean reanalyses are used to provide this baseline. However, the sparsity of the observations results in somewhat differing ocean states appearing equally plausible[21]. As such, understanding the ocean processes occurring in these hindcasts and their sensitivity to the nature of the reanalyses is crucial in understanding the reliability of these decadal predictions.

In this study, we use two assimilation simulations (reanalyses), as well as hindcast simulations from fifteen decadal prediction systems, shown in Table 1. The reanalyses we use are the European Centre for Medium Range Weather Forecasting ORAS4 reanalysis[22] and the UK Met Office Decadal Prediction System 3 assimilation run (denoted DP3-assim)[23], described further in Methods. For clarity, the hindcasts from this latter system are denoted as DePreSys3 in the text. In addition, we use data from long-term preindustrial control simulations conducted with the same climate models. We combine these data to investigate the systematic links between climate model biases, the ensuing prediction systems and uncertainties in the ocean reanalyses. We show that, in the Labrador Sea, despite various methods of initialisation, the prediction systems continue to behave in a similar manner to the comparator unforced control simulations. As a result, the skill of these prediction systems in key metrics such as the AMOC is strongly dependent on whether the biases in the underlying climate models happen to produce similar behaviour to the chosen verifying analysis. It remains an open question at this stage to what extent these persistent biases affect the results of multi-decadal/centennial climate projections.

## Results

**Labrador Sea T/S.** To set the scene in Fig. 1, we show the annual mean of the top 500 m depth averaged temperature and salinity in the Labrador Sea in the two reanalyses and fifteen decadal prediction systems. For clarity, the hindcasts are shown as ensemble means with all valid years averaged together (i.e., the average of all hindcasts available for that year), which may include multiple start dates. The raw data are shown in Supplementary Fig. 1.

With the exception of bcc-csm1-1, all hindcasts and reanalyses indicate warming of the region in the late 1990s and early twentieth century (Fig. 1a). In addition, it can be seen that the DP3-assim reanalysis is consistently warmer than the ORAS4 reanalysis, which may reflect the different assimilation strategies. For example, the spreading of information into the poorly observed (on decadal timescales) boundary current regions may manifest differently between the two systems. Nonetheless, despite the differences in mean temperatures, the two reanalyses give similar annual and decadal variability, which will be the focus of this study. The annual correlation between the two reanalyses over the period is $r = 0.91$, and both reanalyses have similar annual standard deviations (DP3-assim s.d. = 0.35 K, ORAS4 s.d. = 0.34 K).

Considering instead salinity averaged over the same volume (Fig. 1b), there is reduced agreement between the two reanalyses.

**Table 1 The decadal prediction hindcast systems and preindustrial control simulations used in this analysis using the CMIP5 nomenclature**

| Institute (CMIP5 name) | Hindcast model (CMIP5 name) | No. start dates | No. ensemble members | Control model (CMIP5 name) | Initialisation method[a] |
|---|---|---|---|---|---|
| BCC | bcc-csm1-1 | 47 | 3 | bcc-csm1-1 | FF |
| CCCma | CanCM4 | 52 | 3 | CanESM2 | FF |
| CMCC | CMCC-CM | 9 | 1 | CMCC-CM | FF |
| CNRM-CERFACS | CNRM-CM5 | 10 | 1 | CNRM-CM5 | FF |
| ICHEC | EC-EARTH | 10 | 3 | EC-EARTH | FF |
| IPSL | IPSL-CM5A-LR | 10 | 3 | IPSL-CM5A-LR | Anom |
| LASG-CESS | FGOALS-g2 | 10 | 3 | FGOALS-g2 | FF |
| LASG-IAP | FGOALS-s2 | 10 | 3 | FGOALS-s2 | FF |
| MIROC | MIROC4h | 10 | 1 | MIROC4h | Anom |
| MIROC | MIROC5 | 51 | 3 | MIROC5 | Anom |
| MOHC | DePreSys3 | 22 | 10 | HadGEM3-GC2 | FF |
| MPI-M | MPI-ESM-LR | 51 | 3 | MPI-ESM-LR | Anom |
| MPI-M | MPI-ESM-MR | 11 | 3 | MPI-ESM-MR | Anom |
| MRI | MRI-CGCM3 | 11 | 1 | MRI-CGCM3 | Anom |
| NOAA-GFDL | GFDL-CM2p1 | 52 | 3 | GFDL-CM2p1 | FF |

[a]Whether the initialisation method is full-field (FF) or anomaly (anom) is also shown. For specific details of the initialisation methodology, the reader is referred to Table 1. of ref. [36]. For further details of the CMIP5 models and institutions, the reader is referred to Table 9.A.1 of ref. [37] and references therein

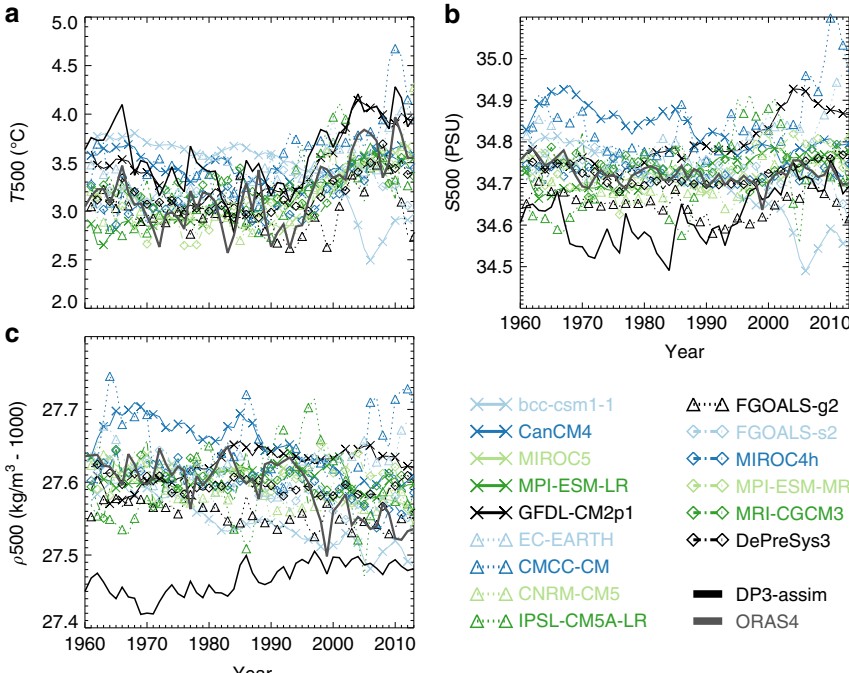

**Fig. 1** Evolution of key quantities in the Labrador Sea. Time series of the volume averaged temperature ($T500$, **a**), salinity ($S500$, **b**), and density ($\rho500$, **c**) in the Labrador Sea (45–60°W, 55–65°N) from the top 500 m in reanalyses and hindcast systems. For visual clarity, the hindcast systems are shown as grand ensemble means across all ensemble member and valid lead times for a given year with symbols and line styles as shown in the legend (identical in all figures). All hindcasts have been lead-time dependent bias corrected and mean-adjusted to compare to the ORAS4 reanalysis over the period from 1960–2013. The reanalyses are shown as solid lines without symbols

This may be related to the relatively sparse observational record of salinity before the twenty first century in this region[24], which is particularly pronounced between 1975 and 1995 in our study region. When confronted with sparse observations, DP3-assim appears to induce much larger annual/decadal variability than ORAS4, and shows a late 1990s increase similar to the temperature record. A key feature of DP3-assim is the use of cross-covariances between temperature and salinity (whereby temperature observations can influence salinity increments and vice versa), which provides a potential mechanism for the input of this higher amplitude salinity variability—compared to ORAS4, in which salinity relaxes to climatology with a timescale of 1 year (see Methods). It is difficult to completely assess whether the real ocean did or did not experience such variability and assessing that is not the focus of this study. Nonetheless, within our study region, subsampling the reanalyses where there exists quality-controlled observational data, suggests that DP3-assim is more consistent with these observations than ORAS4, particularly prior to the ARGO period (see Supplementary Fig. 2). However, it remains a largely open question whether either of the reanalyses can be said to be more plausible in terms of their dynamical evolution.

Some of the hindcast simulations exhibit large decadal salinity variability even after lead-time dependent bias correction, as in DP3-assim, whereas others exhibit little variability similar to ORAS4. For completeness, the correlation skill in T500 and S500 in this region is formally assessed in Supplementary Fig. 3. There is positive T500 skill at all lead times up to 5 years when assessed against either of the reanalyses in all hindcasts, except bcc-csm1-1. The skill in S500 depends more heavily on the choice of truth (reanalysis) against which one compares. In general, increased skill against one reanalysis (e.g., DP3-assim) precludes increased skill against the other reanalysis (e.g., ORAS4). Importantly, the difference in skill when choosing different (though similarly

plausible) reanalyses is also manifested in the dynamically important density ($\rho500$) over the same region.

**Labrador Sea density drivers**. In Fig. 2, we show the correlation skill in density (over the same region as previously) assessed against both ORAS4 (Fig. 2a) and DP3-assim (Fig. 2b). In general, there are more prediction systems that show good skill against ORAS4 than against DP3-assim. This preference for ORAS4 includes hindcasts made with DePreSys3, which has previously been documented[25]. Both of the reanalyses are based on the NEMO[26] ocean model. Five of the prediction systems also use versions of, or precursors to, NEMO (CMCC-CM, CNRM-CM5, EC-EARTH, IPSL-CM5A-LR and DePreSys3), but show no systematic preference for either DP3-assim or ORAS4 (with a similar null result for models not based on NEMO). In addition, although not controlling for differing model systems, there is no systematic relationship between whether the hindcast systems are anomaly or full-field initialised and the subsequent skill in density (or T500 or S500) against either ORAS4 or DP3-assim.

Without knowing which of DP3-assim or ORAS4 is closer to the real world, we cannot say which prediction systems are likely to be more useful in making predictions about the real world future. Note that, the reanalyses agree well during the recent well-observed decade, in part due to the ARGO network[27], but that the skill of multi-annual prediction systems is necessarily estimated over many decades[20]. As such, it is important to begin to understand the mechanisms by which these different estimates of the real world arise, both in the reanalyses and hindcasts, in order to understand these uncertainties and their impacts.

To understand the lead time dependence of density skill against either of the reanalyses, we also show the lead-time dependent density drivers in all the hindcasts. This follows the same method as already used for DePreSys3[25], and is summarised in Methods.

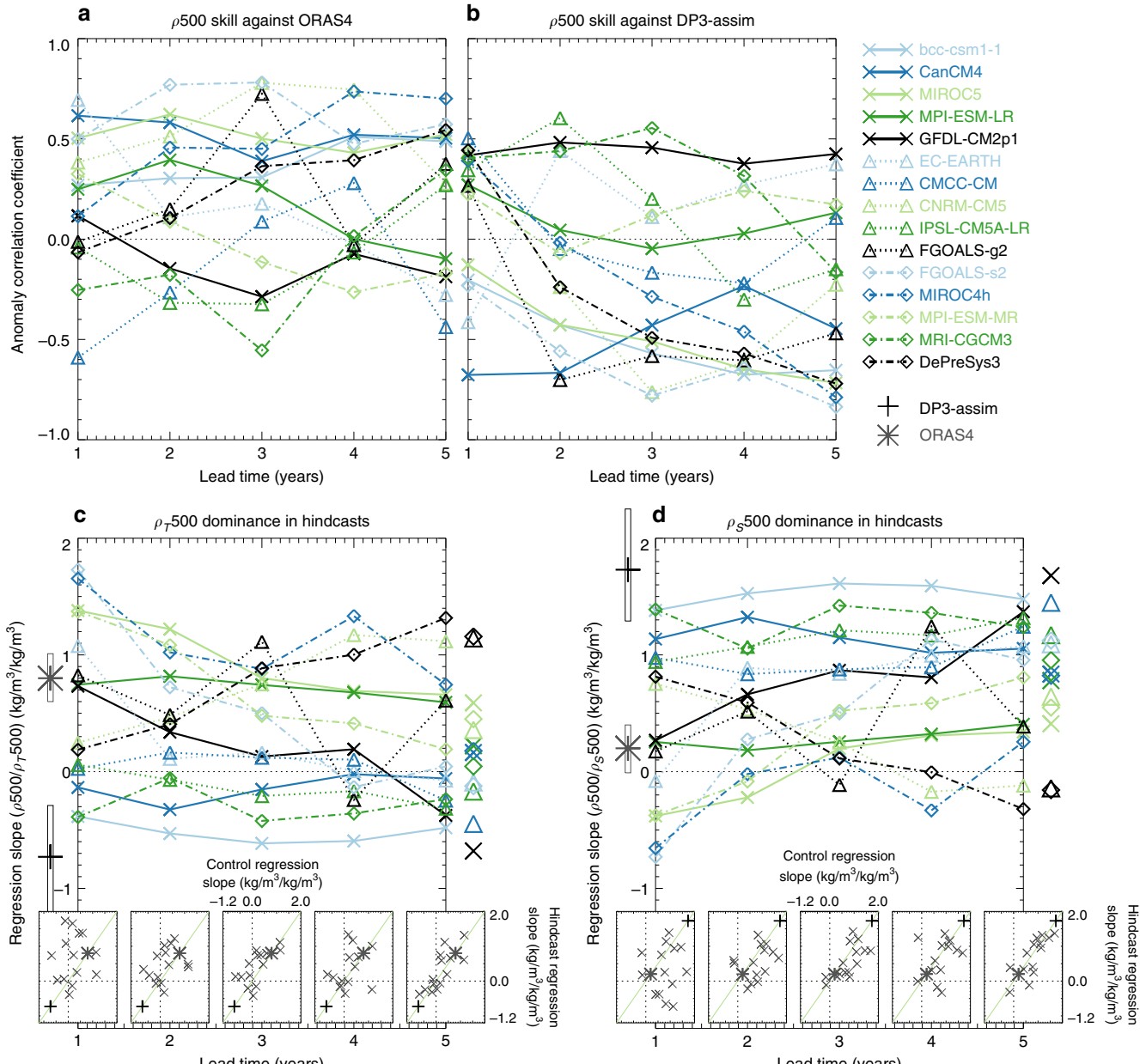

**Fig. 2** The evolution of correlation skill and density drivers in the hindcasts. The correlation skill in the Labrador Sea top 500 m density ($\rho$500) between the hindcast systems and the reanalyses ORAS4 (**a**) and DP3-assim (**b**), as a function of lead time. A lead-time dependent bias correction (assessed against ORAS4 and DP3-assim separately) is applied to the hindcasts before calculating the skill. The regression slope between $\rho$500 and $\rho_T$500 (**c**), or $\rho$500 and $\rho_S$500 (**d**) in the hindcast systems as a function of lead time. For comparison, the same regression slope in ORAS4 (grey star) and DP3-assim (black cross) is displayed to the left of the axes with 90% confidence intervals (boxes), estimated via a bootstrap analysis. In addition, the same regression slope calculated from the control simulations (see Table 1) is shown to the right of the axes (further investigation of the stationarity of the control regression coefficients is shown in Supplementary Fig. 4). Finally, scatter plots of control simulation regression slope (x-axis, constant) against hindcast regression slope (y-axis) are inlaid as a function of lead time to highlight how the hindcasts revert to the nature described by the control simulations within a few years. The reanalyses are also shown with symbols coloured as before and can be seen to span a large portion of the hindcast behaviour. The one-to-one line is shown in green

In general, an increase in temperature-driven density variability ($\rho_T$500, Fig. 2c) mirrors a decrease in salinity-driven density variability ($\rho_S$500, Fig. 2d) due to the approximate linearity of the calculation, but we show both panels for clarity.

The initialisation strategies differ between prediction systems and for many the concept of interannual variability in their associated assimilation is not well defined (for example, in situations where temperature and salinity are instantaneously relaxed into the subsurface ocean only at initialisation[28]).

However, for DP3-assim and the ORAS4 assimilation, there does exist a continuous consistent ocean state. As such, we have also plotted the regression slopes between $\rho$500 and $\rho_T$500, or $\rho_S$500 throughout the reanalyses. These are shown to the left of the axes. It is immediately apparent that these reanalyses are fundamentally different in their assessment of what drives interannual density variability, be it either salinity (DP3-assim) or temperature (ORAS4). Note that, qualitatively similar results are obtained by using a smaller region within the central Labrador Sea, over

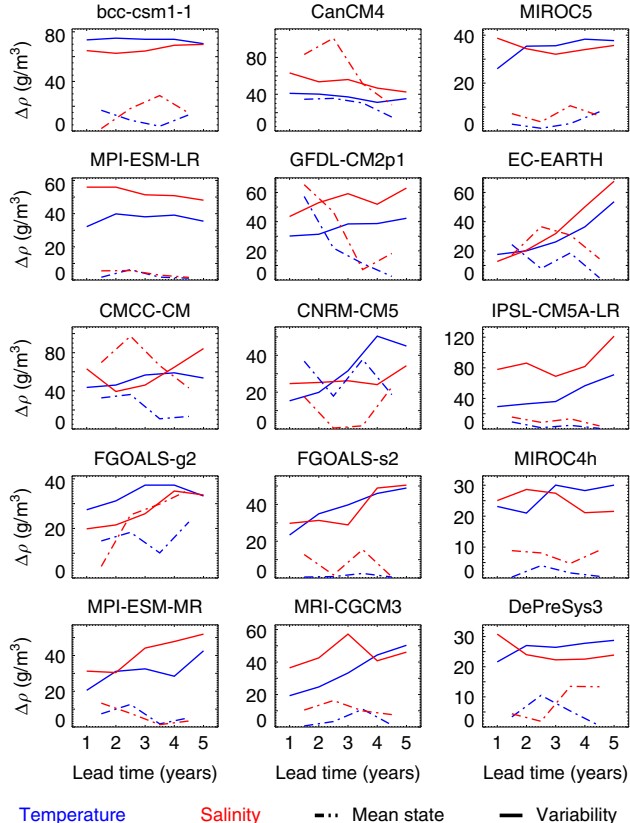

**Fig. 3** The relative roles of the mean state and the variability. Contributions from changes in the mean state (dot-dashed) or changes in the variability (solid) from temperature (blue) and salinity (red) to characteristic density changes in the hindcasts as a function of lead time

just the top 200 m (Supplementary Fig. 5), but we use the present definition for consistency with earlier work. In the hindcasts, some can be said to remain close to one or other reanalysis estimate of the driver of density variability, whereas others prefer to switch within a relatively short timescale. For example, the DePreSys3 prediction system moves away from its own reanalysis (DP3-assim), whereas GFDL-CM2.1 and FGOALS-s2 move towards DP3-assim and away from ORAS4.

This leads to the question: Why do these drifts occur (or not occur), and is there any systematic explanation that can cover the seemingly different behaviour across the 15 hindcast prediction systems? Such an understanding would be valuable in determining what aspects of both prediction systems and reanalyses are most important for creating reliable predictions of the real world North Atlantic Ocean.

**Links between hindcasts and control simulations**. To investigate this question, we have additionally analysed the preindustrial control simulations from the same models as used in the hindcast prediction systems, as detailed in Table 1. In general, prediction systems use an underlying coupled atmosphere-ocean climate model that has also been used to conduct long-term control simulations with constant preindustrial forcings, with the data also uploaded to the Fifth Coupled Model Intercomparison Project (CMIP5) archive. The density drivers from the control simulations are thus shown to the right of the axes.

Despite the different density drivers in the hindcasts, and that some show lead time dependence that is not apparent in others, it is generally the case that after around 5 years the density drivers in the hindcasts and control simulations agree. That is, across the

15 models available on the CMIP5 archive, the driver of the density variability in the underlying control simulation provides a very strong constraint on the driver in the initialised hindcasts. This is additionally shown by the scatter plots that show the relationship between the control and the hindcast density driver for the various hindcast lead times. Here, it can be seen that the models generally align on the 1:1 line after 5 years. Not all centres provided data after a lead time of 5 years, but for those that did the multimodel correlation did not continue to improve after this time (not shown). In the parameters so far investigated, the two reanalyses we have used span a large portion of the distribution determined by the hindcasts, which provides further evidence of the difficulty in determining which (if any) of the prediction systems is likely to make robust predictions of the future.

We have shown that the driver of the Labrador Sea top 500 m interannual density variability in the prediction systems reverts to the underlying control model within 5 years. Despite this, many of the hindcasts have still not reverted to the mean state temperature and salinity structure of their control simulations within 5 years, which leads us to ask: are the changes in the actual mean state (i.e., drifts/adjustments), or changes in the magnitude of innate temperature and salinity variability responsible for the rapid resetting of density drivers to their control simulation values?

**Changes in mean state versus changes in variability**. In Fig. 3, we show the lead-time dependent characteristic density changes associated with changes in the interannual variability of temperature and salinity or changes in the mean state of temperature and salinity. Details of this procedure are provided in Methods.

It can be seen that, in general, the density changes due to the changing annual temperature and salinity variability are larger than those due to the changes in the mean state (the latter associated with drifts or adjustments that occur as a result of initialisation shock). This may partly be because a large fraction of the adjustment occurs within the first year and so is effectively hidden in our analysis. Nonetheless, it can be seen that at subsequent lead times any continued mean state adjustment does not contribute significantly to the density changes, with the exception of salinity drifts in CanCM4 and CMCC-CM.

For some models, the driver of density changes was seen to switch from salinity to temperature (e.g., CNRM-CM5 and DePreSys3, cf. Fig. 2). In these models, the largest characteristic density changes go from being dominated by salinity variability to dominated by temperature variability (Fig. 3). As such, this switch in density drivers can be attributed to a relative increase in the magnitude of temperature—as compared to salinity—variability.

Similarly, for models that switch from temperature- to salinity-driven density variability (e.g., MPI-ESM-MR, FGOALS-s2, EC-EARTH, cf. Fig. 2) this can again be attributed to relative changes in the magnitude of temperature and salinity variability.

We have shown that the drivers of interannual density variability in the various hindcasts revert to that seen in the control simulations within 5 years, and further that this is most often due to a change in the magnitude of variability rather than the mean state (associated with drifts). We now investigate how these systematic changes affect the skill of the prediction systems.

**Effect on Labrador Sea density skill**. From Fig. 2c, d, it can be seen that prediction systems that are based upon underlying models that have a particular driver of density variability are more likely to show skilful predictions when assessed against a reanalysis that exhibits the same driver of density variability and vice versa. This is quantified in Fig. 4a, b.

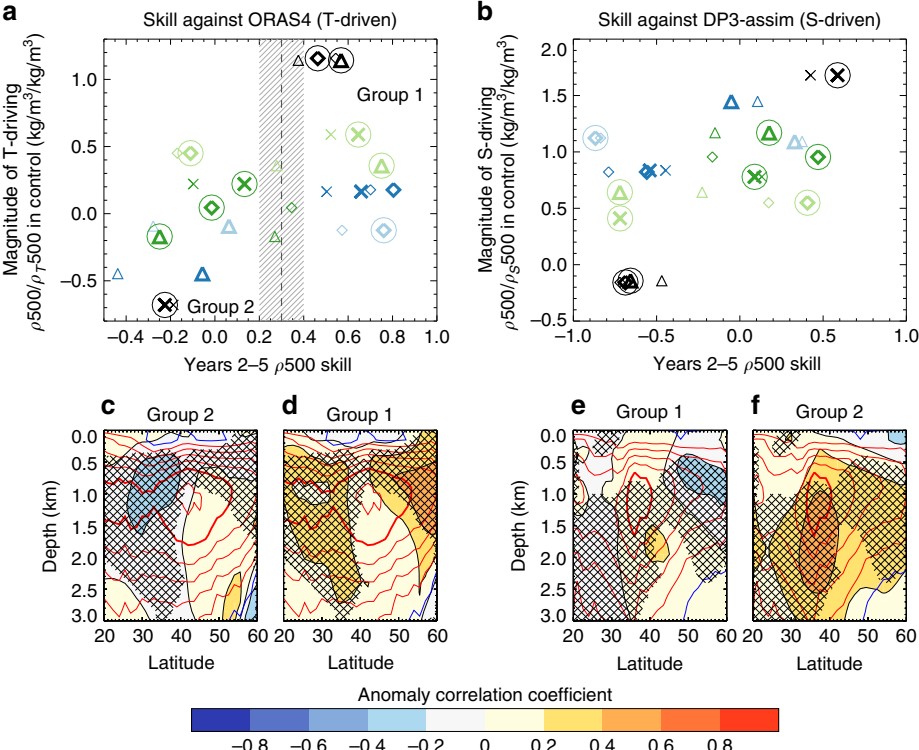

**Fig. 4** From Labrador Sea density drivers to large scale circulation. Correlation skill of years 2 to 5 volume averaged Labrador Sea top 500 m density ($\rho500$) in hindcasts assessed against ORAS4 (**a**) and DP3-assim (**b**) plotted against the regression slope between $\rho500$ and $\rho_T500$ (**a**) or $\rho500$ and $\rho_S500$ (**b**) in the control simulations for the same models. The hindcasts are separated into two groups based on a cut-off of $r = 0.3$ in year 2 to 5 $\rho500$ skill, when assessed against ORAS4. The symbol markers are as in Figs. 1 and 2 and the outlying hindcast system bcc-csm1-1 has been excluded from this analysis. The skill for just year 5 is also shown with the same but smaller symbols. The hindcast systems for which streamfunction data was provided are circled in panels **a** and **b**. The group average years 2 to 5 Atlantic overturning streamfunction skill assessed against ORAS4 (**c**, **d**), and DP3-assim (**e**, **f**), where the groups are as in panels **a** and **b**. Group 1: MIROC5, CNRM-CM5, FGOALS-g2, FGOALS-s2 and DePreSys3. Group 2: MPI-ESM-LR, GFDL-CM2p1, IPSL-CM5A-LR, MPI-ESM-MR and MRI-CGCM3. The time mean streamfunctions in the verifying reanalysis are contoured every three Sverdrups ($1 \times 10^6$ m$^3$ s$^{-1}$) in red (positive, bold at 15 Sverdrups) and blue (negative). A total of 10 hindcast systems provided streamfunction data and every possible 5-member combination of these is computed to assess the significance of these group-average skill estimates. The hatching signifies skill at the 80% level, i.e., the group mean skill falls in either the top or bottom 10% of possible values

There is an approximately linear relationship in either of the panels between the density skill in the hindcasts and the density driver in the control simulations, with ORAS4 finding temperature-driven models to be skilful, whereas DP3-assim finds salinity-driven models to be skilful. That is, it appears to be the underlying nature of density variability (as seen in the control simulations)—and whether this agrees with the verifying reanalysis—that determines a large part of the subsequent skill of a hindcast system.

Nonetheless, although it is generally true that specifying similar density drivers to the target reanalysis leads to higher skill than if that is not the case, there remains plenty of model diversity. This highlights that there are other factors than just the drivers of interannual density variability that lead to skill in Labrador Sea density. Indeed, whether temperature or salinity drives the density variability is not a process but a symptom of actual processes that are represented differently across the models, either due to differing parameterisations or differences in forcings.

We have shown that, in the Labrador Sea, prediction systems generally revert to the density drivers exhibited by the same control simulations within 5 years. This has impacts for skill in the density in the same region, which is a dynamically important quantity. As such, it is appropriate to ask whether these changes have wider impacts, for example affecting the skill in circulation indices outside of the Labrador Sea, such as the AMOC.

## Discussion

To determine the potential for wider impacts, we assess the skill of the AMOC in each hindcast simulation against both the AMOC in DP3-assim and that in ORAS4. We group the models into two groups based on their skill in Labrador Sea density assessed against ORAS4. Models with skill greater than $r > 0.3$ are in Group 1, with the remaining models in Group 2. Using this definition, it also clear that models cluster into the same groups when assessed instead against DP3-assim. Finally, having grouped the models we then show the group-mean skill in the AMOC streamfunction (Fig. 4c–f). The outlying bcc-csm1-1 model was not included (cf. Fig. 1 and Supplementary Fig. 3) and not all centres provided streamfunction data resulting in each group containing 5 models.

It is immediately obvious from this analysis that prediction systems that show good skill in Labrador Sea density variability against a given reanalysis (for the reasons previously outlined) also generally have a much more skilful AMOC when also compared to that reanalysis (Fig. 4d, f). Similarly, choosing the wrong reanalysis to compare against generally results in poor density skill and poor AMOC skill (Fig. 4c, e). Further, despite our density metric focussing on the near surface Labrador Sea, the skill in the AMOC exists throughout the North Atlantic and notably below the 500 m layer, implying that good simulation of the Labrador Sea is important for skill in the large-scale ocean circulation.

Although the various prediction systems yield differing estimates of skill in key variables (e.g., the AMOC), we have provided a framework in which these skill scores can be systematically understood. Skill in the AMOC on timescales beyond the first few years is not solely dependent on the prediction system but also largely depends on which reanalysis is used for validation. More specifically, it depends on whether the control simulation of the forecast model in the prediction system agrees with the reanalysis in key metrics in the Labrador Sea. Further, there is no relationship between the broad initialisation strategy (be it either Anomaly or Full Field initialisation) and either density skill in the Labrador Sea or the AMOC. This is likely because neither strategy can correct for the drifts in variability that are important here.

We have described a strong link between control simulations and prediction systems built upon the same climate models. Climate projections of the next century also use these same models. To what extent are multi-model climate projections of the North Atlantic and beyond systematically dependent on the innate northern North Atlantic density variability within these models?

## Methods
The various systems including the number of ensemble members and other key features are detailed in Table 1. All analysis uses the ensemble mean and annual mean data, unless otherwise stated. The reader is directed to the appropriate papers for description of the DP3-assim reanalysis[23] and ORAS4 reanalysis[22], though some key details are repeated here.

**DP3-assim.** The DP3-assim[23] nudges full-field values of potential temperature and salinity from the Met Office Statistical Ocean Reanalysis (MOSORA)[29] into the HadGEM3-GC2 model with a relaxation time scale of 10 days. Temperature and winds in the atmosphere (6 h relaxation) as well as sea-ice (1 day relaxation) are also nudged in a similar manner. MOSORA uses optimal interpolation and is created through an iterative procedure. For the first iteration a set of global covariances are computed from control integrations of nine different versions of the third Hadley Centre Coupled Model (HadCM3) with different physics parametrizations[30]. These covariances are used to make the first analysis. Two further iterations are made, each using the analysis from the previous iteration, instead of the model control integrations, to make its covariances. In this way, the covariances are influenced by observations when they exist. As there are more temperature than salinity observations in the early period, covariances between temperature and salinity are used to help determine the salinity.

**ORAS4.** ORAS4 uses the NEMO model (version 3.0) and is based on 3D-variational assimilation (NEMOVAR)[22]. It uses observations of potential temperature, salinity and sea-surface height (SSH). The ocean model is forced with ERA-40/ERA-Interim[31]. There is a strong relaxation to observation-based sea surface temperature products, and freshwater flux is also adjusted using constraints from altimeter data and through a relaxation to a monthly climatology from WOA05[32] with a time scale of one year. A change of variables is performed within the assimilation to (assumed) uncorrelated control variables of temperature, unbalanced salinity and unbalanced SSH. Linearised versions of the balance relationships described in refs. [33,34] are used in NEMOVAR to convert from these variables to the total salinity and total SSH, and vice-versa. See ref. [35] for more details. A model bias correction scheme is used in ORAS4. The bias estimate is based on time-accumulated temperature and salinity increments (with a three month relaxation time scale). The bias correction is applied to temperature and salinity directly in the extra tropics and via the pressure gradient (pressure correction) in the tropics.

**Density breakdown.** We follow the method of ref. [25], specifically the 'absolute densities' method as in general we do not have access to year zero/assimilation data. We break density variability down into components arising from separately temperature and salinity variability by keeping one of either salinity or temperature constant (at its time mean value) in the equation of state. This results in the quantities $\rho_T 500$ (density due to temperature variability) and $\rho_S 500$ (density due to salinity variability), anomalies in which add almost linearly to give anomalies in $\rho 500$. We thus regress $\rho_T 500$ and $\rho_S 500$ separately against $\rho 500$ to estimate the relative importance of $\rho_T 500$ and $\rho_S 500$ (and thus temperature and salinity) variability to the actual density variability. This is done as a function of lead time and shown in Fig. 2c, d.

**Characteristic density changes due to mean and variability.** The contribution of mean state changes are estimated by averaging together the lead-time dependent

bias over all start dates for a given lead time and prediction system, and then finding the difference (in temperature and salinity) between one year and the next. Essentially, this estimates the contribution to density from model drift, which can be seen in the requirement of a lead-time dependent bias correction (specifically a bias correction against ORAS4, but similar results are obtained for the correction required against DP3-assim, not shown). This results in, for example, a temperature change ($\pm\Delta T$) valid midway between year 1 and year 2. This is then combined with a reference temperature and salinity ($T_{ref} = 3.5$ K, $S_{ref} = 34.5$ PSU) to estimate a characteristic density change. The same method is used to calculate the characteristic density change associated with the mean state salinity change ($\pm\Delta S$).

Similarly, to estimate the contribution of changes in annual temperature and salinity variability for each prediction system, we find the standard deviation of the lead-time dependent bias over all start dates for each lead time. This results in, for example, a characteristic temperature change ($\pm\Delta T$) valid at year 1. We proceed as above to calculate characteristic density changes, but note that as we are not taking the difference between years, the validity time of these density changes (associated with variability) is offset by half a year, compared to the density changes associated with the mean state (above).

**Code availability.** The code to analyse the climate model data is available from the authors upon reasonable request.

**Data availability.** The control and hindcast simulations are available on the CMIP5 archive and the reanalyses from http://icdc.cen.uni-hamburg.de/1/projekte/easy-init.html (accessed July 2017).

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

## Acknowledgements

We were supported by the Joint UK BEIS/Defra Met Office Hadley Centre Climate Programme (GA01101). L.H. was additionally supported by the DYNAMOC project, which is funded by the RAPID-AMOC programme. We would like to thank Dan Lea of the Met Office and Jon Robson of the University of Reading for useful conversations during the preparation of this manuscript.

## Author contributions

M.B.M. and L.H. jointly designed the analysis idea and scope. The analysis was conducted by M.B.M. Both authors contributed to the writing of the paper.

## Additional information

**Competing interests:** The authors declare no competing interests.

