## [Peer Review File · Nature Communications]

Reviewers' comments:

Reviewer #1 (Remarks to the Author):

This work is innovative and worth publication. It highlights the limitations of skill in the North Atlantic. This work is very rigorous and highlights the limitations of the skill compared with two ocean reanalysis. In those reanalysis, the density metrics in the Labrador sea is either temperature or salinity driven and thus the skill of the climate model or decadal predictions model is largely or not dependent on those.

I will have few comments on the article.

Line 43 and Table 1: You mention the two main types of predictions, which have the bias, removed in anomaly or FF method. Do the types of data or reanalysis or model used for initialization be important to?

Line 73: Do you have some idea why the bcc-csm1-1 does not show this warming?

Line 98-100: Do you have some explanation for this? Is it model dependent? Or from the initialization method?

Line 113-117: In this work, two reanalysis are investigated which are both based on the NEMO ocean model. Maybe it could have been interesting to use also one using another ocean model such as those describe in Karspeck et al. (2015). You mention 5 predictions systems that use the NEMO model do you have similar conclusion with the prediction system using other ocean model?

Line 125: How many years of good reanalysis do we need or a necessary for a skillful multi-annual predictions system?

Line 126: Maybe not in this study but maybe from the literature, do you know if some reanalysis are better to represent processes in this region?

Line 143-146: Maybe it could be nice to have a spread of this behavior with a large range of ocean reanalysis.

In the conclusion, do you think you will get better agreement and skill if you look at a more integrated quantity such as the MLD or volume of dense water formed in the Labrador sea?

Reviewer #2 (Remarks to the Author):

Limits on determining the skill of the North Atlantic Ocean decadal predictions

The authors investigate the mechanism of predictability in the Labrador Sea in 15 state-of-the-art climate prediction systems and two widely used reanalysis (ORAS4 and DP3). In particular they try to isolate the variable and the time scale responsible for the density change in the region that is known to influence the Atlantic Meridional Overturning Circulation. The period of study covers the period 1960-2010. Both reanalysis and hindcasts (exception made of one) show similar evolution

for the heat content but there are large disagreement in the variability of the salt content. Observations are crucially lacking there and it is unclear which is correct. In one of the reanalysis the density changes are mainly driven by temperature variability while it is controlled by salinity variability in the other one. Such disagreement is also found in the hindcasts. Once the impact of initialisation dissipates, the controlling variable in the hindcast may revert (in the 2-5 year lead time) to its preferred mechanism of variability (estimated by looking to the pre-industrial simulation of the model). The variable controlling the density change have implication on the pattern of the meridional overturning circulation. The manuscript is interesting and cast new understanding on the potential limitation of current climate projection and prediction systems. It is very well written; entertaining and the figure are of high quality and comprehensive. However, some parts of the manuscript are unclear and would need further investigation/clarification. I would recommend that the authors address the comments below before the manuscript be accepted for publication.

L18 I disagree with the statement that the key metrics are independent of the details of the prediction system. Here the only 2 aspects of a prediction system area investigated: which model is used and whether full field or anomaly initialisation is used. Some other detail can be equally or more important (see comment below).

Sometime the authors use the acronym NASPG (L29) some other times NA SPG (L33).

L 52: There is no such thing as perfect observation. Observations will always have some degree of uncertainty.

L41: There may be earlier reference to that statement ?

- Keenlyside, N. S., et al. "Advancing decadal-scale climate prediction in the North Atlantic sector." *Nature* 453.7191 (2008): 84.
- Robson, J. I., R. T. Sutton, and D. M. Smith. "Initialized decadal predictions of the rapid warming of the North Atlantic Ocean in the mid 1990s." *Geophysical Research Letters* 39.19 (2012).
- Yeager, Stephen, et al. "A decadal prediction case study: Late twentieth-century North Atlantic Ocean heat content." *Journal of Climate* 25.15 (2012): 5173-5189.

L71, please clarify what all-valid year means. Do you mean " the average of all hindcast available at that year?"

Table 1: Considering the information provided in Table 1 as an exhaustive description of the details of a prediction system is wrong. While the choice of the model or FF/Anom are important one, there are many other choice that are equally important. The parametrisation or the running of a model can have very large influence. Also on the initialisation side : Are a system assimilating data or are they initialised from a reanalysis performed with a different system. If data assimilation method is used, what method is used and what observations are assimilated ?

Fig 1. A lot of the analysis in the manuscript is based anomaly correlation of rho500. Correlation can be sometime misleading. Please show the time series of rho500 (e.g. in Fig 1) for the hindcast and reference reanalysis.

On Fig2:

- Why did the author use the control simulations with pre-industrial forcing and not the historical forcing simulation? Which of T or S drive the density change can evolve with global warming.
- Is the period of calculation for the regression comparable to that of the hindcast ?
- Following on the first comment of Fig 2, one may expect that the driving variable for density

change depends on the mean temperature of the area. Have the authors attempted to relate the driving variable that to the temperature bias of each system in the region ? Temperature bias are extremely stable ? There is also a relatively strong temperature difference between ORAs4 and DP3.

- Can the authors show the histogram of the regression value calculated over a running 50-year windows from the long PI simulation for each model in the supplementary material. I would be curious to know if some models transit from a temperature driven density to salinity driven density (or vise-versa) with time.
- Some more details:
 - o The authors should estimate confidence interval for ORAS4 and DP3 regression estimate (L143).
 - o The correlation of DP3 with ORAS4 and vice versa should be shown. It is very hard to see the difference between DP3 and ORAS4, I would recommend using a different symbol and if possible a color that is more different.

L90 Some of the details of DP3 assimilation are given here. If so it should be explained why the system differ from ORAS4. "DP3 assim assimilation" is redundant. Cross covariance is unclear (covariance are always cross). Does the author means "Use of covariance to estimate the multivariate updates".

L98 here the ensemble mean is used. With a predominance of temperature observations, one merely expect the ensemble spread of salinity to be larger and the ensemble mean flatter. Does the ensemble averaging influence the variability of salinity ? if so, maybe the median would be better suited ?

L125 This is likely because there are sufficient data to controlled the system during the ARGO period (see for example Zhang, S., A. Rosati, and T. Delworth. "The adequacy of observing systems in monitoring the Atlantic meridional overturning circulation and North Atlantic climate." *Journal of Climate* 23.19 (2010): 5311-5324.)

L140 Remove the extra coma

L 169 do not use contraction for is not.

L204 Can you explain the behavior of FGOALS-g2 where the mean drift seems to increase over time ?

Section 5 is relatively confusing. Up to now we look at the bias-corrected hindcast and now we look at the impact of the density change w.r.t to year-to-year variability ? The ranges of density change between models are very different (on Figure 3) and for most models both temperature and salinity seems equally important. The authors should try to improve the clarity of this section.

L204 Are not MIROC5 and MIROC 4h belonging to that category as well ? They do not show such behavior in Fig 2 ?

L213 MPI-ESM did seem to switch.

L242 add a coma after the Labrador Sea ?

In Figure 4, it seems that MIROC 4h belongs to group1

L299 is the relaxation time scale the same for the atmosphere and sea ice ?

L 302 Does the ensemble evolves from the previous assimilation cycle or are they initialized on the

first iteration from the previous analysis mean + perturbations? It seems that the method is what is standardly call "bred vector data assimilation method" if so mention it and add a reference. Please indicate the localization radius (vertically and horizontally).

L313 It is not surprising that ORAS4 underestimate salinity variability prior to the ARGO period because they uses a 1-year relaxation to climatology.

L 343 Please clarify the meaning of the sentence starting by "Specifically ..."

In Supplementary Figure 2, it would be most valuable if the number of observations available with time were added. If there are times with good observation covered (for example a hydrographic mission) in the Labrador Sea; the value could be reported on Fig 1.

Reviewer #3 (Remarks to the Author):

This paper aims to look at determining the skill of decadal predictability in the northern North Atlantic Ocean. This is an important region because of its links to the large scale Atlantic overturning circulation. These linkages also mean there is potential for predictability with up to a decade lead time. The authors examine this question using a large suite of CMIP5 models, as well as two ocean reanalysis products. The authors show that the predictive skill is not a functional of the predictive system but instead depends on the reanalysis, and more specifically how the inter-annual density variations are set in the underlying climate model and chosen reanalysis.

This is an interesting paper, important for those wanting to use climate models to make decadal predictions. It brings up important issues related to bias and sensitivity that then will need to be further explored by the community. The paper is well written and the figure quality is reasonable. However, I do have a couple of major concerns that I think would need to be addressed by the authors before this manuscript is suitable for publications. These concerns, as well as more minor points, are detailed below.

The authors analyze the Labrador Sea, and define their study region in the caption to figure 1. Why do the authors pick a region that combines both the interior and the boundary currents, given those regions have very different properties and dynamics? Why not use an isobaths/interior based definition like many studies to allow for a focus on the interior region? Otherwise contrasting signals in the two regions might be cancelling each other out. Furthermore why the choice of the top 500 m for vertical depth averaging. Most observational studies generally consider ~200 m as the boundary between the fresher colder upper layer and the saltier and warmer layer below containing Irminger Water. The surface layer also strongly responds to the air-sea fluxes. See Straneo (2006) for a good example of such a discussion. Thus I worry that signals will be masked by the use of a 500 m layer that averages through watermasses with very different properties.

Additionally, in multiple places, the authors make statements like "poorly observed boundary current regions", "variability which is by definition unobserved", etc. Although more data is always more desirable, I think these statements are far too extreme. There are multiple annual sections that have been taken across both boundary currents, at least in summer, for longer than a decade. The main AR7W section has an even longer record. Long term mooring records exist at 53N. Argo floats have now been around for 15+ years. Additionally, past major experiments (such as the Labrador Sea Convection Experiments, the weather ships, etc.) mean that there are periods with extensive data that then allows some level of analysis of long term variability in the region. See recent papers by Yashayaev et al for example, which show mapped estimates of interior T and S variability back to the 1940s. Given that, I find the lack of an attempt to at least evaluate which reanalysis might be closer to the real world as a major limitation. I'm not arguing there is enough data that there might not be large potential errors. But to just wave it away as poorly unobserved without trying to examine the issue in detail is not correct either.

Additional Points:

Line 46 – the northwest corner is a name used for a specific region where the North Atlantic Current retroflects, and isn't thus the same as the Labrador Sea. I'd thus suggest using a different choice of words for describing the location of the Labrador Sea in the sub-polar gyre.

References – many issues with lack of capitalization

Authors' comments:

We thank all three reviewers for their time in reviewing our work. These comments have proved useful in improving our manuscript. A point by point response to your comments is provided below - original comments in black and responses in red.

Note that all reviewer line numbers refer to the original manuscript. All response line numbers (red) refer to the Tracked Changes version of the updated manuscript.

Reviewers' comments:

Reviewer #1 (Remarks to the Author):

This work is innovative and worth publication. It highlights the limitations of skill in the North Atlantic. This work is very rigorous and highlights the limitations of the skill compared with two ocean reanalysis. In those reanalysis, the density metrics in the Labrador sea is either temperature or salinity driven and thus the skill of the climate model or decadal predictions model is largely or not dependent on those.

Thank you for reviewing our work!

I will have few comments on the article.

Line 43 and Table 1: You mention the two main types of predictions, which have the bias, removed in anomaly or FF method. Do the types of data or reanalysis or model used for initialization be important to?

We find no systematic relationships but it is hard to make broad groupings based on other factors - the ones we highlight in Table 1 are already the broadest ways to characterise decadal prediction systems. Indeed, it is this difficulty that inspires us to approach the problem from a process-based standpoint (i.e. the driver of density variability). We have added a reference to the table where the reader can find further details of the systems investigated here.

Line 73: Do you have some idea why the bcc-csm1-1 does not show this warming?

Further analysis suggests that the Labrador Sea in bcc-csm1-1 becomes increasingly disconnected from the wider subpolar gyre throughout the twentieth century. Instead, it appears to co-vary more strongly with the East Greenland Current, which is cooler and fresher than the wider SPG and would explain the cooling/freshening that is seen in bcc-csm1-1.

Line 98-100: Do you have some explanation for this? Is it model dependent? Or from the initialization method?

This is due to the underlying model, as noted in the discussion of Figure 2. It does not appear to be due to the initialisation methodology, at least in the broadest FF-versus-Anom sense.

Line 113-117: In this work, two reanalysis are investigated which are both based on the NEMO ocean model. Maybe it could have been interesting to use also one using another ocean model such as those described in Karspeck et al. (2015). You mention 5 prediction systems that use the NEMO model do you have similar conclusion with the prediction system using other ocean model?

We attempted to get hold of further full 3D monthly temperature and salinity data for the period 1960-2013+ for other reanalyses but were not able to procure it. However, as our analysis focuses on the drivers of density variability, and we have two reanalyses with opposing drivers, we feel this is sufficient to investigate the hindcasts in this context. Note that the focus of our study is the prediction systems (hindcasts) and not the reanalyses themselves. On the second point, we again find no systematic relationship (note added to main text). L134-135

Line 125: How many years of good reanalysis do we need or a necessary for a skillful multi-annual predictions system?

Many decades according to other studies, which we have now cited. L143.

Line 126: Maybe not in this study but maybe from the literature, do you know if some reanalysis are better to represent processes in this region?

We have investigated this somewhat in Supplementary Figure 2, which suggests that DP3-assim is a closer fit to the available observations than ORAS4. However, a more thorough analysis would need to be done to account for the uncertainties involved. L104-109.

Line 143-146: Maybe it could be nice to have a spread of this behavior with a large range of ocean reanalysis.

As above, this would be nice but is outside the scope of this paper. We hope our results (for the prediction systems) will inspire others to analyse the same metrics in the ocean reanalyses.

In the conclusion, do you think you will get better agreement and skill if you look at a more integrated quantity such as the MLD or volume of dense water formed in the Labrador sea?

Previous work has highlighted the very large diversity in MLDs in CMIP5 models (Menary and Wood, 2017; Heuze, 2017) and that MLDs can drift very rapidly in a prediction system (Menary and Hermanson, 2016) so it's not clear that such a metric would lead to improved agreement between the models. In addition, many modelling centres did not upload mixed layer diagnostics to the CMIP archive, which would make such an analysis very difficult.

Reviewer #2 (Remarks to the Author):

Limits on determining the skill of the North Atlantic Ocean decadal predictions

The authors investigate the mechanism of predictability in the Labrador Sea in 15 state-of-the-art climate prediction systems and two widely used reanalysis (ORAS4 and DP3). In particular they try to isolate the variable and the time scale responsible for the density change in the region that is known to influence the Atlantic Meridional Overturning

Circulation. The period of study covers the period 1960-2010. Both reanalysis and hindcasts (exception made of one) show similar evolution for the heat content but there are large disagreement in the variability of the salt content. Observations are crucially lacking there and it is unclear which is correct. In one of the reanalysis the density changes are mainly driven by temperature variability while it is controlled by salinity variability in the other one. Such disagreement is also found in the hindcasts. Once the impact of initialisation dissipates, the controlling variable in the hindcast may revert (in the 2-5 year lead time) to its preferred mechanism of variability (estimated by looking to the pre-industrial simulation of the model). The variable controlling the density change have implication on the pattern of the meridional overturning circulation. The manuscript is interesting and cast new understanding on the potential limitation of current climate projection and prediction systems. It is very well written; entertaining and the figure are of high quality and comprehensive. However, some parts of the manuscript are unclear and would need further investigation/clarification. I would recommend that the authors address the comments below before the manuscript be accepted for publication.

Thank you for the nice comments!

L18 I disagree with the statement that the key metrics are independent of the details of the prediction system. Here the only 2 aspects of a prediction system are investigated: which model is used and whether full field or anomaly initialisation is used. Some other detail can be equally or more important (see comment below).

We have rephrased this to highlight we are looking at some of the details of the prediction system. Nonetheless, the broad initialisation method and underlying climate model are undoubtedly some of the first order details. L19-20

Sometime the authors use the acronym NASPG (L29) some other times NA SPG (L33).

This has been fixed.

L 52: There is no such thing as perfect observation. Observations will always have some degree of uncertainty.

This has been rephrased. L54

L41: There may be earlier reference to that statement ?

- Keenlyside, N. S., et al. "Advancing decadal-scale climate prediction in the North Atlantic sector." *Nature* 453.7191 (2008): 84.
- Robson, J. I., R. T. Sutton, and D. M. Smith. "Initialized decadal predictions of the rapid warming of the North Atlantic Ocean in the mid 1990s." *Geophysical Research Letters* 39.19 (2012).
- Yeager, Stephen, et al. "A decadal prediction case study: Late twentieth-century North Atlantic Ocean heat content." *Journal of Climate* 25.15 (2012): 5173-5189.

If you mean Original Line 41, then the current reference predates these. If you mean Original Line 43 then the reference we use is appropriate for tropical storms, which we specifically mention.

L71, please clarify what all-valid year means. Do you mean " the average of all hindcast available at that year?"

Yes we do. This has been clarified. L76-77

Table 1: Considering the information provided in Table 1 as an exhaustive description of the details of a prediction system is wrong. While the choice of the model or FF/Anom are important one, there are many other choice that are equally important. The parametrisation or the running of a model can have very large influence. Also on the initialisation side : Are a system assimilating data or are they initialised from a reanalysis performed with a different system. If data assimilation method is used, what method is used and what observations are assimilated?

Similar to the comment for L18, we are merely highlighting some of the first order measures of the systems and do not intend for the table to be exhaustive. We have added text to refer the reader to where they can find more detailed information (see Table caption). It is also important to note that we are not specifically testing FF v Anom (for example) but are testing *emergent properties* of the decadal prediction systems. We show the details in Table 1 to orient the reader and to highlight that we are aware of the first order descriptors of the systems.

Fig 1. A lot of the analysis in the manuscript is based anomaly correlation of rho500. Correlation can be sometime misleading. Please show the time series of rho500 (e.g. in Fig 1) for the hindcast and reference reanalysis.

This has been added to Supplementary Figure 1.

On Fig2:

- Why did the author use the control simulations with pre-industrial forcing and not the historical forcing simulation? Which of T or S drive the density change can evolve with global warming.

We use control simulations in order to characterise the underlying preferred nature of the models, in the absence of any forcings. Nonetheless, other appropriate measures would be a twentieth century control, or the transient simulations, as suggested. Nonetheless, for the period 1960-2013, we find no systematic changes in the historical simulations (not shown).

- Is the period of calculation for the regression comparable to that of the hindcast ?

It uses the full period, but we have now added a supplementary figure (Figure S4) investigating the stationarity of this relationship (see also below).

- Following on the first comment of Fig 2, one may expect that the driving variable for density change depends on the mean temperature of the area. Have the authors attempted to relate the driving variable that to the temperature bias of each system in the region ? Temperature bias are extremely stable ? There is also a relatively strong temperature difference between ORAs4 and DP3.

We tested this prior even to testing the relationship with variability. We find no systematic relationship with the mean state. For anomaly-assimilation methods, the models are already at/around their mean state. For full-field assimilation, this is likely because, although the models drift after initialisation, they do not reach their preferred (i.e. control) mean state within 5 years, and in fact are generally still not close to it. This is related to the fact that the

control mean states are actually quite well separated (see Figure 1 of “Exploring the impact of CMIP5 model biases...” <http://onlinelibrary.wiley.com/doi/10.1002/2015GL064360/full>)

- Can the authors show the histogram of the regression value calculated over a running 50-year windows from the long PI simulation for each model in the supplementary material. I would be curious to know if some models transit from a temperature driven density to salinity driven density (or vise-versa) with time.

We have added this as requested (Figure S4).

- Some more details:

- o The authors should estimate confidence interval for ORAS4 and DP3 regression estimate (L143).

We have added this as requested (Figure 2 and caption)

- o The correlation of DP3 with ORAS4 and vice versa should be shown. It is very hard to see the difference between DP3 and ORAS4, I would recommend using a different symbol and if possible a color that is more different.

The (lack of) correlation can now be inferred from Supplementary Figure 2. We have changed the symbols so as to make this figure clearer.

L90 Some of the details of DP3 assimilation are given here. If so it should be explained why the system differ from ORAS4. “DP3 assim assimilation” is redundant. Cross covariance is unclear (covariance are always cross). Does the author means “Use of covariance to estimate the multivariate updates”.

We have fixed “DP3-assim assimilation”. We mean that temperature is allowed to co-vary with salinity, rather than just with temperature. This has been clarified. L97-98

L98 here the ensemble mean is used. With a predominance of temperature observations, one merely expect the ensemble spread of salinity to be larger and the ensemble mean flatter. Does the ensemble averaging influence the variability of salinity ? if so, maybe the median would be better suited ?

We recomputed Figure 1 and 2 using just the first ensemble member and found little change. Note that the hindcasts are free-running and so although the temperature initialisation may be better (i.e. closer to reality) it does not necessarily follow that the ensemble member evolution will be smaller in temperature than in salinity.

L125 This is likely because there are sufficient data to controlled the system during the ARGO period (see for example Zhang, S., A. Rosati, and T. Delworth. "The adequacy of observing systems in monitoring the Atlantic meridional overturning circulation and North Atlantic climate." *Journal of Climate* 23.19 (2010): 5311-5324.)

Thanks. We have added this. L101-102

L140 Remove the extra coma

Done.

L 169 do not use contraction for is not.

This has been changed.

L204 Can you explain the behavior of FGOALS-g2 where the mean drift seems to increase over time ?

This implies that the salinity (or rather the salinity component of the linearised density EOS) is showing an exponential drift with time. Given the importance of convection in this region, it is possible that the model is systematically initialised in a state not conducive to convection and that this spins up over time, with a feedback on to salinity. Unfortunately, there is no MLD data provided in order to test this hypothesis. However, we also note that the FGOALS-g2 has a strong AMOC when free-running (“Oceanic climatology in the coupled model FGOALS-g2: Improvements and biases” Lin et al. 2013) driven by strong mixing, which is consistent with the hypothesis above.

Section 5 is relatively confusing. Up to now we look at the bias-corrected hindcast and now we look at the impact of the density change w.r.t to year-to-year variability ? The ranges of density change between models are very different (on Figure 3) and for most models both temperature and salinity seems equally important. The authors should try to improve the clarity of this section.

- For Figure 3, this is largely an artefact of showing mean state and variability changes on the same scales. Nonetheless, comparison of Figure 3 with either Figure 2 or 4 is consistent. Note also that the “density-driver” can show small but important changes in this context: temperature and salinity anomalies generally co-vary in this region such that they are warm/saline or cool/fresh. As such, a small change in the characteristic density changes associated with T/S can have a very large impact if one now becomes bigger than the other.
- For the general clarity of the section, we have reworded this section.

L204 Are not MIROC5 and MIROC 4h belonging to that category as well ? They do not show such behavior in Fig 2 ?

The MIROC models both show a shift to increasingly temperature-driven density variability (Figure 3). This is consistent with Fig 2 panels c and/or d, in which the MIROC models show increasingly temperature-driven (and decreasingly salinity-driven) density variability, even if not a complete sign change.

L213 MPI-ESM did seem to switch.

Similar to the above, in Figure 3 MPI-ESM-MR shows increasingly *salinity*-driven density variability, although there isn't a sign change here (i.e. a crossing of the solid red and blue lines). In Figure 2 (c and d), an increase in salinity-driven density variability is again seen, although here the sign does change. Nonetheless, in both figures the tendency is the same and consistent. In addition, it is the power of multimodel analyses such as ours that the overall signal-to-noise is improved compared to single models in which there are undoubtedly other important processes also at play.

L242 add a comma after the Labrador Sea ?

This has been done.

In Figure 4, it seems that MIROC 4h belongs to group1

It does but there was no AMOC data on the CMIP5 archive for this model and so it doesn't contribute to the streamfunction plots. Only models that uploaded streamfunction data are used in Figure 4c-f, which fortunately results in an equal number of models in each group (5 in each).

L299 is the relaxation time scale the same for the atmosphere and sea ice ?

No it isn't - this has been clarified. L323-324

L 302 Does the ensemble evolves from the previous assimilation cycle or are they initialized on the first iteration from the previous analysis mean + perturbations? It seems that the method is what is standardly call "bred vector data assimilation method" if so mention it and add a reference. Please indicate the localization radius (vertically and horizontally).

It is not a bred vector method. We have attempted to clarify our description further.

L313 It is not surprising that ORAS4 underestimate salinity variability prior to the ARGO period because they uses a 1-year relaxation to climatology.

We have now added a comment on this in the main text. L101-102

L 343 Please clarify the meaning of the sentence starting by "Specifically ..."

Lead time dependent bias correction for a hindcast must be done against a baseline reanalysis. Here, we mean that we have done this against ORAS4, but that one could use DP3-assim and get very similar results. This has been clarified. L372

In Supplementary Figure 2, it would be most valuable if the number of observations available with time were added. If there are times with good observation covered (for example a hydrographic mission) in the Labrador Sea; the value could be reported on Fig 1.

We have added this information to supplementary Figure 2 as requested. We thought Figure 1 was busy enough as it is, but instead we have added mention of the times of best observational coverage in the main text. L94

Reviewer #3 (Remarks to the Author):

This paper aims to look at determining the skill of decadal predictability in the northern North Atlantic Ocean. This is an important region because of its links to the large scale Atlantic overturning circulation. These linkages also mean there is potential for predictability with up to a decade lead time. The authors examine this question using a large suite of CMIP5 models, as well as two ocean reanalysis products. The authors show that the predictive skill is not a functional of the predictive system but instead depends on the reanalysis, and more specifically how the inter-annual density variations are set in the underlying climate model and chosen reanalysis.

This is an interesting paper, important for those wanting to use climate models to make decadal predictions. It brings up important issues related to bias and sensitivity that then will need to be further explored by the community. The paper is well written and the figure quality is reasonable. However, I do have a couple of major concerns that I think would need to be

addressed by the authors before this manuscript is suitable for publications. These concerns, as well as more minor points, are detailed below.

We are pleased that you found the paper interesting and appreciate your time in reviewing it.

The authors analyze the Labrador Sea, and define their study region in the caption to figure 1. Why do the authors pick a region that combines both the interior and the boundary currents, given those regions have very different properties and dynamics? Why not use an isobaths/interior based definition like many studies to allow for a focus on the interior region? Otherwise contrasting signals in the two regions might be cancelling each other out. Furthermore why the choice of the top 500 m for vertical depth averaging. Most observational studies generally consider ~200 m as the boundary between the fresher colder upper layer and the saltier and warmer layer below containing Irminger Water. The surface layer also strongly responds to the air-sea fluxes. See Straneo (2006) for a good example of such a discussion. Thus I worry that signals will be masked by the use of a 500 m layer that averages through watermasses with very different properties.

We used this region because it has been used on several previous occasions in the literature, in particular in the current line of work (Menary et al., “Exploring the impact of CMIP5...”; Menary, Hermanson, Dunstone, “The impact of Labrador Sea...”). In addition, in the latter paper we tested the effect of using just the Central Labrador Sea, as defined by Yashayaev & Loder (2016) and found it made little difference to the relationships between density and either the salinity or temperature components. Nonetheless, to further test the effect of both the use of an interior region and a more restricted depth range we have recomputed the density are just its temperature and salinity components on comparable figures for both the original region top 500m (top) and Central Labrador Sea top 200m (bottom) in both reanalyses (Supplementary Figure 5). The figure legends highlight the correlation between the density (black lines) and its components (salinity component in red, temperature component in blue). When switching from the original (top) to interior (bottom) definitions it can be seen that DP3-assim remains strongly salinity driven (high correlation with ρ_S). Likewise, ORAS4 remains temperature-driven (higher correlation with ρ_T than with ρ_S) but the correlation is slightly weakened by the different box choice. As such, the general result (namely that DP3-assim is primarily salinity driven and ORAS4 is primarily temperature driven) remains. L104-112. L164-176.

Additionally, in multiple places, the authors make statements like “poorly observed boundary current regions”, “variability which is by definition unobserved”, etc. Although more data is always more desirable, I think these statements are far too extreme. There are multiple annual sections that have been taken across both boundary currents, at least in summer, for longer than a decade. The main AR7W section has an even longer record. Long term mooring records exist at 53N. Argo floats have now been around for 15+ years. Additionally, past major experiments (such as the Labrador Sea Convection Experiments, the weather ships, etc.) mean that there are periods with extensive data that then allows some level of analysis of long term variability in the region. See recent papers by Yashayaev et al for example, which show mapped estimates of interior T and S variability back to the 1940s. Given that, I find the lack of an attempt to at least evaluate which reanalysis might be

closer to the real world as a major limitation. I'm not arguing there is enough data that there might not be large potential errors. But to just wave it away as poorly unobserved without trying to examine the issue in detail is not correct either.

1. We have toned down the statements on the paucity of observations. Our aim was not to denigrate the observational network but merely to note that the Labrador Sea is a harder region to make measurements in than either other parts of the ocean or land/atmosphere. In addition, as noted above, in our previous work we made a comparison between our present definition of the Labrador Sea and the interior definition of Yashayaev & Loder and found little difference in the reanalyses. L54. L84. L92. L104. L138
2. In Supplementary Figure 2 we have subsampled the raw (quality controlled) observational profiles to compare against the two reanalyses. We have also added information on the observational density to this figure. In addition, we have added further comment on which (if either) of the reanalyses performs better against the available observations. We find that DP3-assim is a better fit against the observations and have now mentioned this specifically. A more in-depth comparison between the reanalyses and available observations in this region would be worthwhile but would be a whole paper in itself. L104-109
3. We have added an additional supplementary figure (Supplementary Figure 5) that explores the effect in the reanalyses of subsampling to a more interior and nearer-surface region. This is discussed in response to the point above. L164-167

Additional Points:

Line 46 – the northwest corner is a name used for a specific region where the North Atlantic Current retroflects, and isn't thus the same as the Labrador Sea. I'd thus suggest using a different choice of words for describing the location of the Labrador Sea in the sub-polar gyre.

This has been changed.

References – many issues with lack of capitalization

These have been fixed

Reviewers' comments:

Reviewer #2 (Remarks to the Author):

Limits on determining the skill of the North Atlantic Ocean decadal predictions

It is the second review iteration of the manuscript. While the authors have addressed some of my concerns, I still disagree with some of the statements. Furthermore figures in the supplementary material may have highlighted a major concern in the dynamical feasibility of one of the reanalysis product (DP3 assim). In my view the manuscript is still not ready for publication, but has the potential to make a nice contribution to Nature Communications.

I still disagree with the broad statement starting L 13: "Skill in key metrics ..., are largely independent of details of the prediction system such as model or initialisation."

In my view the authors show the contrary. First, two reanalyses products using a similar model version but different initialisation strategy (MOSORA and NEMOVAR) suggest that density variability is controlled by a different state variable. The difference seems to be routed in the details of the covariance matrix of the data assimilation methods (so in the initialisation). Second, I do not consider all models that are based on the NEMO community model to be the same model. Using a different sea ice compartment or using different parameters changes completely the behaviour of the model. This is nicely exemplified by your figure 4 of the supplementary material, where all NEMO based models depict very different behaviour. In this sense key metric are thus not independent of details in the models.

The Supplementary Figure 5 shows the density variability of the two reanalyses. I find it extremely worrying that the variability in DP3-assim is mostly explained by the density-increasing trend. There is no sign of the notorious 1995 SPG shift. Density changes there are supported by observations (e.g. Van Aken et al. 2011 using tracer and temperature and salinity observations), altimetry data, proposed mechanism of variability (Yeager and Robson 2017 for a review on that topic). This merely suggests that the choices in MOSORA for the covariance between T and S are not suitable for the region studied. It was for instance advanced in Counillon et al. 2016 that in the Labrador Sea, the covariance between T and S changes with the phase of the SPG. I acknowledge that finding which of the reanalysis is most realistic is not the main scope of the paper, but in this case it seems that DP3-ASSIM is suspicious, which is important for the following understanding of the paper.

It is said that the reanalysis agrees well in the recent well observed period (L 129) but they are not for density. I strongly insist that the density evolution of the two reanalyses is shown in the main manuscript, since most of the following figure are based on that time series.

- van Aken, Hendrik M., M. Femke De Jong, and Igor Yashayaev. "Decadal and multi-decadal variability of Labrador Sea Water in the north-western North Atlantic Ocean derived from tracer distributions: Heat budget, ventilation, and advection." *Deep Sea Research Part I: Oceanographic Research Papers* 58.5 (2011): 505-523.
- Yeager, S. G., and J. I. Robson. "Recent Progress in Understanding and Predicting Atlantic Decadal Climate Variability." *Current Climate Change Reports* 3.2 (2017): 112-127.
- Counillon, François, et al. "Flow-dependent assimilation of sea surface temperature in isopycnal coordinates with the Norwegian Climate Prediction Model." *Tellus A: Dynamic Meteorology and Oceanography* 68.1 (2016): 32437.

I agree that salinity observations are lacking making it is hard to identify which is more realistic. What about altimetry data? Which of the reanalyses and prediction systems (temperature driven or salinity driven ones) shows best skill in explaining the variability of the sea surface height in the NA subpolar gyre region.

There is an error in the way citation display in the text ("author?" is appearing). See for example in the label of Table 1; L 325, 326 328, 334 etc ...

Line 80 units are missing.

Could the axis range added for the regression of Figure 2

L 194, When referring to "control simulations"; are the authors referring to the pre-industrial simulation ? If so this is rather obvious because of global warming. Does the hindcasts revert to their historical simulation?

Sentence starting L 286 is confusing, could it be re-phrase ?

Line 279: This statement is not entirely correct because a prediction system consists of a model and its initialisation strategy. Here the hindcasts revert because the preferred variability of the model. The reversal is also not immediate; it takes < 5 years. I agree that : "Predictions systems revert to their innate preferred variability mode within less than 5 years."

Reviewer #3 (Remarks to the Author):

The authors have done a good job revising their manuscript. I only have a couple of minor comments at this stage.

Line 50 - LSW contributes to the upper part of lower limb of the AMOC. The authors should be clear with their sentence here.

Line 83 - I'd prefer less well observed to poorly observed.

Line 154-158 - There seems to be an issue with this sentence.

Reviewers' comments:

Reviewer #2 (Remarks to the Author):

Limits on determining the skill of the North Atlantic Ocean decadal predictions

It is the second review iteration of the manuscript. While the authors have addressed some of my concerns, I still disagree with some of the statements. Furthermore figures in the supplementary material may have highlighted a major concern in the dynamical feasibility of one of the reanalysis products (DP3 assim). In my view the manuscript is still not ready for publication, but has the potential to make a nice contribution to Nature Communications.

I still disagree with the broad statement starting L 13: "Skill in key metrics ..., are largely independent of details of the prediction system such as model or initialisation."

In my view the authors show the contrary. First, two reanalysis products using a similar model version but different initialisation strategy (MOSORA and NEMOVAR) suggest that density variability is controlled by a different state variable. The difference seems to be rooted in the details of the covariance matrix of the data assimilation methods (so in the initialisation). Second, I do not consider all models that are based on the NEMO community model to be the same model. Using a different sea ice compartment or using different parameters changes completely the behaviour of the model. This is nicely exemplified by your figure 4 of the supplementary material, where all NEMO based models depict very different behaviour. In this sense key metrics are thus not independent of details in the models.

"I do not consider all models that are based on the NEMO community model to be the same model" - we agree. Indeed, this is exactly our point - prediction systems based on the same ocean submodel do not always trend towards ORAS4. Their behaviour is "independent" of the ocean submodel but depends on an emergent property that is their T/S variability.

We find ourselves agreeing entirely with your statements in the above paragraph despite having reached opposite conclusions.

Part of the confusion may be that when we say "skill" at any point in the manuscript we are referring to the skill in the hindcasts as measured against these reanalyses, NOT the skill of the reanalyses against independent observations. The latter would be an entirely different paper.

Most importantly, in our analysis, *it does not matter* what the initialisation of the hindcasts is. All that is important are two things: 1) That a multi-year free-running hindcast is conducted, and 2) that an unforced, long control simulation using the same/similar model has also been conducted.

Essentially, we then compare these two sets of simulations and find that the skill of the hindcast system (assessed against reanalysis) depends not on details of the hindcast system but on whether its control simulation looks similar to the reanalysis to which one compares.

If we have understood your argument correctly, it is (very briefly):

1. Skill depends on the reanalysis chosen (and whether the control is like this)
2. The reanalysis can also be called the initialisation
3. Thus skill depends on the initialisation

However, it is the link between 2 and 3 that does not hold. The reanalyses only act as the initialisation for a few of the models, and as previously noted, even in those (e.g. DP3-assim and DePreSys3 hindcasts) they can result in poor skill when both are compared.

The Supplementary Figure 5 shows the density variability of the two reanalyses. I find it extremely worrying that the variability in DP3-assim is mostly explained by the density-increasing trend. There is no sign of the notorious 1995 SPG shift. Density changes there are supported by observations (e.g. Van Aken et al. 2011 using tracer and temperature and salinity observations), altimetry data, proposed mechanism of variability (Yeager and Robson 2017 for a review on that topic). This merely suggests that the choices in MOSORA for the covariance between T and S are not suitable for the region studied. It was for instance advanced in Counillon et al. 2016 that in the Labrador Sea, the covariance between T and S changes with the phase of the SPG. I acknowledge that finding which of the reanalysis is most realistic is not the main scope of the paper, but in this case it seems that DP3-ASSIM is suspicious, which is important for the following understanding of the paper. Differences in trends in this location depend greatly on the specifics of the location chosen, such as the depth range and horizontal extent (cf. papers cited below). As such, we are not surprised that the density evolution in our Figures does not precisely match those in different regions. In response to the previous reviews, we added supplementary figures exploring this sensitivity (Supp Fig 2 and Supp Fig 5). Nonetheless, what is most important is that all of our work with 1) reanalyses, 2) hindcasts, and 3) control simulations are self consistent in using the same location.

On the specific question of whether DP3-assim should be considered suspect in this region: We have extended Supp Figure 2 to include now salinity, temperature, and density. Our analysis suggests that when using independent observations in precisely the same region, rather than being suspect, DP3-assim appears to reproduce these subsampled observations much better than ORAS4. That is - density changes that are supported by observations look a lot like DP3-assim, when using the same definitions of depth range and horizontal extent.

It is said that the reanalysis agrees well in the recent well observed period (L 129) but they are not for density. I strongly insist that the density evolution of the two reanalyses is shown in the main manuscript, since most of the following figure are based on that time series.

We have now included the density evolution in Figure 1. Note that even if one or other of the reanalyses show an trend in density, this is removed during the lead-time dependent bias correction (when assessing the skill of the hindcasts).

- van Aken, Hendrik M., M. Femke De Jong, and Igor Yashayaev. "Decadal and multi-decadal variability of Labrador Sea Water in the north-western North Atlantic Ocean derived from tracer distributions: Heat budget, ventilation, and advection." *Deep Sea Research Part I: Oceanographic Research Papers* 58.5 (2011): 505-523.
- Yeager, S. G., and J. I. Robson. "Recent Progress in Understanding and Predicting Atlantic Decadal Climate Variability." *Current Climate Change Reports* 3.2 (2017): 112-127.
- Counillon, François, et al. "Flow-dependent assimilation of sea surface temperature in isopycnal coordinates with the Norwegian Climate Prediction Model." *Tellus A: Dynamic Meteorology and Oceanography* 68.1 (2016): 32437.

I agree that salinity observations are lacking making it is hard to identify which is more realistic. What about altimetry data? Which of the reanalyses and prediction systems (temperature driven or salinity driven ones) shows best skill in explaining the variability of the sea surface height in the NA subpolar gyre region.

We feel the present manuscript is already long enough and as previously stated and acknowledged - assessing which reanalysis produce is more realistic is not the goal of this paper. We note that DP3-assim does not assimilate SSH data (but the skill has been assessed in Roberts et al. "On the Drivers and Predictability of Seasonal-to-Interannual Variations in Regional Sea Level" (2015)). ORAS4 does assimilate SSH. As such a comparison of the two reanalysis products would be somewhat unfair. We have previously assessed the skill of the DePreSys3 hindcasts against the DP3-assim in dynamic sea level in the same region and found that this followed the skill in density, i.e. it reverted to the underlying model. The mechanisms were the same as described in the present paper. "The impact of Labrador Sea temperature and salinity variability on density and the subpolar AMOC in a decadal prediction system" (2016).

There is an error in the way citation display in the text ("author?" is appearing). See for example in the label of Table 1; L 325, 326 328, 334 etc ...

Author has been left in when the citation is part of the sentence

Line 80 units are missing.

These have been added. L80

Could the axis range added for the regression of Figure 2

The figure is already quite busy - but we have added a note on the range in the caption. The ranges are fixed and the same in all the inlaid panels.

L 194, When referring to “control simulations”; are the authors referring to the pre-industrial simulation? If so this is rather obvious because of global warming. Does the hindcasts revert to their historical simulation?

Yes, we are referring to the PI Control simulations. The difference between the PI Control simulations and the historical simulations is much smaller than the difference between either the PI Control (or historical) and the hindcasts/reanalyses. This may be because the historical simulations are initialised from the PI Controls but the ocean state has not changed much and has thus retained the underlying drifts/biases. As such, the results are insensitive to the choice of baseline (control or historical). We choose PI Control as the baseline because there are no transient forcings to consider and because we are using it to understand the underlying nature of the models.

Sentence starting L 286 is confusing, could it be re-phrased ?

This has been rephrased. L291-293

Line 279: This statement is not entirely correct because a prediction system consists of a model and its initialisation strategy. Here the hindcasts revert because the preferred variability of the model. The reversal is also not immediate; it takes < 5 years. I agree that : “Predictions systems revert to their innate preferred variability mode within less than 5 years.”

We agree that a prediction system includes both the initialisation strategy and the submodels. Despite the differing initialisation strategies, we find no link between them and the skill. As such, the skill is independent of the initialisation strategy (for the metrics we have investigated and for the broad definitions of strategy (i.e. FF or Anom) we have considered). Nonetheless, we have slightly rephrased this sentence. L279-286. See also our general response, above.

Reviewer #3 (Remarks to the Author):

The authors have done a good job revising their manuscript. I only have a couple of minor comments at this stage.

Line 50 - LSW contributes to the upper part of lower limb of the AMOC. The authors should be clear with their sentence here.

This has been clarified. L43

Line 83 - I'd prefer less well observed to poorly observed.

This has been changed. L83

Line 154-158 - There seems to be an issue with this sentence.

This has been fixed.

Reviewers' comments:

Reviewer #2 (Remarks to the Author):

I am sorry to say that I am not satisfied with the answer provided. Again, I really like the paper and I think it contains original material that is valuable for the community, but I disagree with some statements (for example L251, L280), which are nicely summarised by the sentence in the abstract.

L 13: "Skill in key metrics ..., are largely independent of details of the prediction system such as model or initialisation."

First, this statement is too vague. What is the meaning of model, initialisation, skill (matching the reanalysis used initially) and at which time scale this statement is correct. It is important because such statement can very easily be misinterpreted.

Second and more problematic, I do not feel that sufficient evidence has been shown to hold this statement.

- The skill in matching either of the two reanalysis depends largely on the choice of the model used for running the hindcasts. Beyond 5 year lead time, hindcasts have reverted to the mechanism of variability dictated by the model selected; (mechanism depicted in PI simulation).
- The two reanalysis converges to different mechanism of variability. Their variability is controlled by a mix of the model behaviour, observations (sampling and variability) as well as assumption in the data assimilation method. Here, the two reanalyses differ in their choice of data assimilation leading to divergence in the mechanisms driving the density variability in the SPG region.
- Regarding the role of initialisation (as mean by the authors; i.e. FF or Anomaly), there does not seems to be sufficient evidence to demonstrate that skill is largely independent of the choice of initialisation. There are no clean comparisons (see Table 1) to hold such statement and all FF and anomaly hindcasts are done with a different model, which has been shown to have a large influence. However, I agree that it seems to have comparatively less influence than the choice of the model beyond few years lead time.

It would be good to mentioned the name and version of the model used for ORAS4 (L319).

Author comments

We sincerely thank the reviewer for their time in reviewing this manuscript. We hope we have addressed their concerns below.

Reviewer line numbers refer to the reviewed manuscript version, our response line numbers refer to the new version showing markup (red).

Matthew Menary and Leon Hermanson

Reviewer #2 (Remarks to the Author):

I am sorry to say that I am not satisfied with the answer provided. Again, I really like the paper and I think it contains original material that is valuable for the community, but I disagree with some statements (for example L251, L280), which are nicely summarised by the sentence in the abstract.

L 13: “Skill in key metrics... are largely independent of details of the prediction system such as model or initialisation.”

We are sorry not to have answered this as clearly as required. Upon further reflection and discussion, we have attempted to clarify our position again. It appears that we are using “independent of the model” in an essentially opposite sense to the way it is being understood here, and indeed perhaps we are wrong. The key point of the paper is this:

- (Multiannual) skill in Lab Sea predictions depends on the “truth” one uses. Whether you end up with good (or bad) skill depends on whether the preferred mechanisms of variability in your model are similar (good) or dissimilar (bad) to the chosen reanalysis, where the reanalyses suggest two essentially opposing and equally plausible mechanisms.

In this sense the skill could, we accept, be said to be a function of the model used. Our interpretation of the above is that the skill is a function of the “agreement in a particular characteristic (density variability) between the reanalysis and the model” and not the perceived *quality* of the prediction model. i.e. It doesn't matter which model you pick *as long as you pick a verifying reanalysis that behaves in the same way*.

It may also be worth highlighting that the DePreSys3 predictions are more skilful when assessed against the ORAS4 reanalysis than against the DePreSys3 reanalysis – for the reasons stated above.

Nonetheless, “independent of the initialisation” on Line 251 is not well defined and has been removed.

On Line 282-284, “independent of the prediction system and...” has been changed in light of the discussion above to “not solely dependent on the prediction system but also largely depends...”

First, this statement is too vague. What is the meaning of model, initialisation, skill (matching the reanalysis used initially) and at which time scale this statement is correct. It is important because such statement can very easily be misinterpreted.

We have added detail to this sentence to clarify the timescale, that we are talking of coupled models, and that we mean the broad characteristics of the initialisation (i.e. FF or Anom). We are limited by the word count of the abstract in providing any more specifics, though these are in the main manuscript. (L14 and L16)

Second and more problematic, I do not feel that sufficient evidence has been shown to hold this statement.

- The skill in matching either of the two reanalysis depends largely on the choice of the model used for running the hindcasts. Beyond 5 year lead time, hindcasts have reverted to the mechanism of variability dictated by the model selected; (mechanism depicted in PI simulation).
- The two reanalysis converges to different mechanism of variability. Their variability is controlled by a mix of the model behaviour, observations (sampling and variability) as well as assumption in the data assimilation method. Here, the two reanalyses differ in their choice of data assimilation leading to divergence in the mechanisms driving the density variability in the SPG region.

These are indeed the key points of the paper. The link between the two being that if the control variability in a model system is similar to that in the reanalysis then the predictions will be good. We hope that our addressing of the “independent” issue (above) has been helpful.

- Regarding the role of initialisation (as mean by the authors; i.e. FF or Anomaly), there does not seems to be sufficient evidence to demonstrate that skill is largely independent of the choice of initialisation. There are no clean comparisons (see Table 1) to hold such statement and all FF and anomaly hindcasts are done with a different model, which has been shown to have a large influence. However, I agree that it seems to have comparatively less influence than the choice of the model beyond few years lead time.

We have added a caveat to this sentence (Line 124) to highlight that we are not doing a clean comparison here.

It would be good to mentioned the name and version of the model used for ORAS4 (L319).

This has been added. Line 321.

REVIEWERS' COMMENTS:

Reviewer #2 (Remarks to the Author):

The authors have answered all of my comments in a rigorous manner. The paper is clear and concise. I would like to congratulate the authors for a very nice and useful contribution.

Regards